# Relational Pattern Benchmarking on the Knowledge Graph Link Prediction Task

Afshin Sadeghi[1,2], Hirra Abdul Malik[1], Diego Collarana[2,3], and Jens Lehmann[1,2]

[1]SDA Research Group, University of Bonn, Germany
[2]Fraunhofer IAIS, Dresden, Germany
[3]Universidad Privada Boliviana, Bolivia
*{s6hiabdu,lehmann}*.uni-bonn.de,
*{afshin.sadeghi,diego.collarana.vargas,jens.lehmann}@iais.fraunhofer.de*

## Abstract

Knowledge graphs (KGs) encode facts about the world in a graph data structure where entities, represented as nodes, connect via relationships, acting as edges. KGs are widely used in Machine Learning, e.g., to solve Natural Language Processing based tasks. Despite all the advancements in KGs, they plummet when it comes to completeness. Link Prediction based on KG embeddings targets the sparsity and incompleteness of KGs. Available datasets for Link Prediction do not consider different graph patterns, making it difficult to measure the performance of link prediction models on different KG settings. This paper presents a diverse set of pragmatic datasets to facilitate flexible and problem-tailored Link Prediction and Knowledge Graph Embeddings research. We define graph relational patterns, from being entirely inductive in one set to being transductive in the other. For each dataset, we provide uniform evaluation metrics. We analyze the models over our datasets to compare the model's capabilities on a specific dataset type. Our analysis of datasets over state-of-the-art models provides a better insight into the suitable parameters for each situation, optimizing the KG-embedding-based systems.

**Keywords:** Machine Learning, Knowledge Graphs Embedding, Link Prediction, Benchmarking, Dataset, Relational Pattern, Inductive, Transductive

## 1 Introduction

Knowledge graphs constitute a significant part of NLP since the 70s. However, after the announcement of big hubs such as Google, Facebook, and Microsoft in the 90s, the growth of research in this particular field became evident as states Hogan et al. (2021). The different elements of a knowledge graph are represented in the form of triplets *(h, r, t)* where *(h, t)* represents entities a.k.a. *'nodes'*, whereas *r* tells the relationship between them, which is also known as *'edge'*. According to Yang and Mitchell (2017), NLP has gotten a new scope after the advancements in the field of knowledge graphs, easing the communication with machines. Various applications such as *information retrieval* Xiong et al. (2017), *question answering* Hao et al. (2017) and *recommender systems* Zhang et al. (2016) use knowledge graphs to improve their performance. Some of the most used knowledge graphs include *DBpedia* Lehmann et al. (2014), *Yago* Suchanek et al. (2007), *Freebase* Bollacker et al. (2008) and *WordNet* Miller (1995).

Despite being in demand, KGs still face many issues, such as data incompleteness. To tackle the issue, link prediction models observe the patterns in knowledge graphs based on how facts are connected together. According to Wang et al. (2019), the goal of the link prediction task is to map the entities/relations to low dimensional vectors capturing the structure of the knowledge graph, which

35th Conference on Neural Information Processing Systems (NeurIPS 2021) Track on Datasets and Benchmarks.

helps predict the likelihood score of the triple. Despite advancements in benchmarks, a significant chunk is still unexplored. In this paper, we enhanced the work of Sarkar et al. (2020) by building various datasets on the principles of known and valuable facts using the Freebase and Wordnet datasets, categorizing them into different patterns for benchmarking. The relation of each triplet is observed and then grouped into categories. The categories include different patterns, e.g., symmetry has the 'same' relation between two entities such as 'friends', whereas inverse has two 'different and directed' relations between the entities such as 'father and son'. We make categories of datasets and observe the link prediction (LP) models over them.

With the goal to set up a benchmark that separates the task of testing KG embedding models from the models, we extend the MLwin-Hobbit platform for benchmarking various trained methods, which are implemented in different environments (e.g. PyTorch and Java). This extension is crucial for a reproducible evaluation and fair comparison of methods.

We train state-of-the-art KG embedding models from scratch with our RAW datasets, providing experimental results that give exciting research directions. We consider the most popular and unified evaluation metrics along with the AUC-PR test in all combinations. Our experiments suggest that the link predicting models are scalable to large-scale datasets and graphs. These results indicate fruitful guidance for future research in KG Link Prediction and KG Embeddings.

## 1.1 Our Contribution

We propose several datasets[1] by classifying triplets into their respective classes according to their patterns, keeping in mind the properties from both inductive and transductive types. Therefore, we extract four categories from each class: Fully Inductive, Fully Transductive, CountBased Inductive, and either Head or Tail Inductive. Each category is further divided into patterns of Symmetry, Anti-symmetry, Inverse and Inductive, making a total of **32** datasets. We developed methods for extracting explicitly separated patterns and also made automatic emending methods to avoid data leakage between detests.

The datasets are also designed on the basis of unification to benchmark them onto different link predicting models. A significant setback in the benchmarking of knowledge graph models was observed. Therefore the work done by Hu et al. (2021) has been extended keeping in mind our set of data. A fair comparison is generated to help choose the best model and dataset combination for NLP Research. The previous research on benchmarking datasets was too general, we provide the specific approach to the type of datasets. We explore the characteristics of the datasets that can be potential performance boosters. To sum up our contribution, we created a reproducible evaluation environment that is user-friendly for all. We designed our benchmark datasets keeping the ease of use in mind.

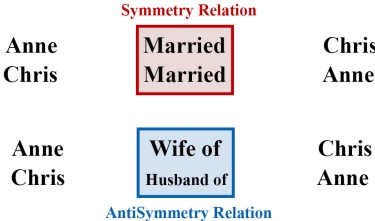

Figure 1: Example of triplet categorization.

## 2   Related Work

**Dataset Building**   We build our datasets keeping in mind the dataset building strategies from OGB (Hu et al. (2021)) as well as the CODEX (Safavi and Koutra (2020)) and TU-Dataset (Morris et al. (2020)). CODEX is gathered from thirteen domains ranging from medicine to music. This dataset is build on the principles of snowball sampling to extract data, while in our study, we searched for

---

[1]All datasets, scripts and extended results are available: `https://github.com/mlwin-de/relational_pattern_benchmarking`

specific relational patterns to enable on pattern-specific evaluations. TU-Dataset, a unified set of over 120 datasets from several domains, targets graph classification and regression tasks, while in our study, we focus on the link prediction task. We use the characteristics visualization technique for the datasets and the required properties for the characteristics analysis, of the aforementioned studies, as a baseline to define our dataset.

**Benchmarking** Benchmarking datasets help to compare and evaluate the LP models, parameters, and procedures as well as the statistics of the different datasets that are evaluated. The benchmarked dataset as described by Rossi et al. (2021) is useful for two basic analyses, efficiency and effectiveness analysis. The dataset of Hu et al. (2021) is kept as our ultimate guidance and standard to support the benchmarking task. CODEX dataset was benchmarked by Safavi and Koutra (2020) with unified evaluation strategies and empirical analysis. Relational Patterns of Inversion, Symmetry, and Composition were studied and subsets were prepared accordingly. We use the same strategy of making sub-datasets and then benchmarking on a number of models. Evaluation technique of using MRR and Hit Ratios was considered but along with the introduction of **AUC-PR**. Using GNN and Graph Kernel methods as used by Morris et al. (2020) gave us a new direction to use Teru et al. (2020) GraIL for our dataset benchmarking. In the study of relation patterns, we include the most frequent patterns. In the related works, they are studies that consider experiments on more complex logical rules, such as Meilicke et al. (2018) that evaluate on Inverse Equivalence and Subsumption rules, and in this direction, Yang et al. (2017) evaluates the performance of the knowledge base inference methods on a dataset of grid paths of different lengths.

In order to perform a link prediction benchmarking based on FAIR principles, we integrated the models in BenchEmbedd (Sadeghi et al. (2021b)), a platform that aims at benchmarking big linked data. Such benchmark experiments are frozen into docker containers, which can be accessed, reproduced, and reused easily with little prior knowledge of the test platform. The system allows researchers to make and test systems without having to worry about standardized hardware. Our trained models and datasets will be publicly available as a BenchEmbedd platform[2] for anyone to use and benchmark systems.

## 3 Dataset Building

Relation Extraction, the sub-field of information extraction, is one of the core techniques that support ML research. It organizes the structural information into groups according to the need (Wang et al. (2021)). We based our study on the two following datasets and extracted subsequent sub-datasets in the form of the stated patterns.

- **FB15K** A freebase dataset with a total of 592,213 triplets with 14,951 entities and 1,345 relationships. This factual dataset contains 483,142 Train triplets, 59,071 Test triplets and 50,000 Valid triplets. It dataset contains many entities from the wiki-link data.
- **WN18** A dataset extracted from Wordnet version 3 with a total of 141,442 triplets with 40,943 entities and 18 relationships. The dataset contains 141,442 Train triplet, 5,000 Test triplets and 5,000 Valid triplets. The dataset supports text analysis and provides with dictionary/thesaurus. Lexical relationship between synsets are stated by this dataset.

### 3.1 Relational Datasets

Pattern extraction is the core task in building datasets for machine learning research. Pattern type suggests the type of link prediction model that works best for the given dataset. Patterns, that are also expressed as rules, each have different suitability to the embedding models. Link Predictors learn the specific pattern of the datasets and then match rule-based patterns to provide reasoning. The patterns we considered to build our datasets are stated below:

### 3.1.1 Symmetry

This relational pattern is a sub-category of **Equivalence** pattern. Therefore, it is a binary relation that works in both directions. The relation can also be stated by the **Equal to** property, for instance, if

---

[2]https://github.com/mlwin-de/BenchEmbedd

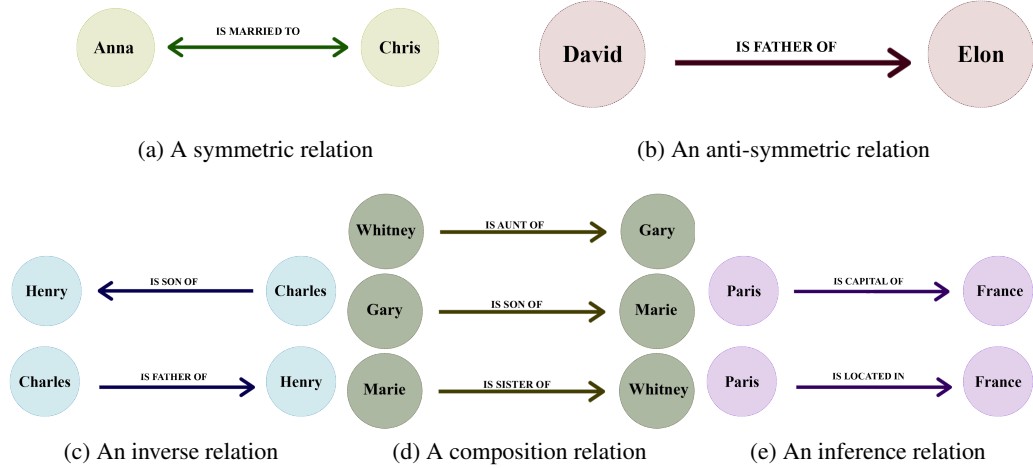

(a) A symmetric relation          (b) An anti-symmetric relation

(c) An inverse relation     (d) A composition relation     (e) An inference relation

Figure 2: We define five different relational patterns to create our datasets and benchmark.

a=b then b=a A relation is symmetric if:

$$\forall a, b \in X (aRb \Leftrightarrow bRa)$$

$$r(a, b) \Rightarrow r(b, a)$$

If $R^T$ represents the converse of R, then R is symmetric if and only if $R = R^T$. Marriage, Friendship, and Partners are a few examples of symmetric relations.

### 3.1.2 Anti-Symmetry

The rule of Anti-Symmetry is opposite to Symmetry. It is a directed rule that states if a relation *R* binds A to B, the same can not work in the opposite direction, binding B to A. The rule is written as

$$r(a, b) \Rightarrow \neg r(b, a)$$

Relations such as Owner (to Tenant), Parent (to Child), and Singer (to Song) are Anti-Symmetric. Figure 2b shows a few examples of this rule.

### 3.1.3 Inverse

It is a binary relation stating two opposite relations for a set of entities. It is possible to assume a unique inverse relation for every relation. Inverse pattern between two set of two triples occurs when they have the opposite relation directions and have the same entities.

$$r_2(x, y) \Rightarrow r_1(y, x)$$

Parent-Child and Teacher-Student are examples of the inverse relations. Figure 2c shows how inverse relation is represented.

### 3.1.4 Composition

This binary relation which is also termed as **relation multiplication** is basically a **compound** relation which states the relation that can not exist without the existence of another relation. For Example, for the relation *Aunt*, the relation of *Sister* and *Son/Daughter* must exist in order to prove someone as an Aunt of somebody. In mathematical terms, relation $r_1$ is composed of relation $r_2$ and relation $r_3$ if:

$$\forall x, y, z : r_2(x, y) \wedge r_3(y, z) \Rightarrow r_1(x, z)$$

Figure 2d states that Paris is the Capital of France, According to the composition property, Paris must be in France to be its capital.

Table 1: Inductive Setting Datasets

| Inductive | | Train | Test | Valid | Total Entities | Total Relations | % of Original Dataset |
|---|---|---|---|---|---|---|---|
| **FB15K** | **Symmetry** | 4254 | 542 | 542 | 3447 | 51 | 0.901 |
| | **Anti-Symmetry** | 12930 | 3494 | 3884 | 8304 | 433 | 3.429 |
| | **Inverse** | 4753 | 2568 | 2568 | 7745 | 641 | 1.670 |
| | **Inference** | 3489 | 2824 | 2745 | 6083 | 611 | 1.530 |
| **WN18** | **Symmetry** | 2322 | 272 | 272 | 4344 | 5 | 1.893 |
| | **AntiSymmetry** | 16650 | 4698 | 4697 | 20552 | 18 | 17.203 |
| | **Inverse** | 8728 | 903 | 904 | 13842 | 17 | 6.958 |
| | **Inference** | 844 | 99 | 99 | 1639 | 15 | 0.688 |

### 3.1.5 Inference

Inference relation pattern is one of the **logical** rules we formed datasets of. The rule states that we can deduce a relationship between two entities from the knowledge of another relationship between the two of them:

$$\forall h, t : (h, r_1, t) \Rightarrow (h, r_2, t)$$

where $h$ and $t$ are entities and $r_1$ and $r_2$ are relations between them. Figure 2e gives an example of inference relation which states Paris is the capital of France and thus, according to r1, Paris much be in France as well. Therefore, *(Paris, isCapitalof, France)* ⇒*(Paris, isLocatedin, France)*

## 3.2 Evaluation Dataset Setting

There exist two different methods for the dataset division into train and test/validation subsets, where the composition of entities of each setting defines the evaluation setting:

**Inductive Setting**   In an inductive setting, the entities during training are not found in the test dataset. The part of entities are kept missing and their relations are made to be found by the LP models. The number of disjoint entities varies in an inductively set dataset, fully disjoint sets are fully inductive and thus difficult for the models to predict.

**Transductive Setting**   A dataset is divided to train and test/validation in a transductive setting when the occurrence of entities is ensured to be in the training procedure if it appears either in a test or valid sets. Transductive set datasets are best for Entity Specific Embedding. All entities in the training set are present in the test set and thus a model has trained embeddings for them specifically.

## 4 A New Set of Pattern Specific Datasets

### 4.1 The Standard Patterns

We extracted a subset of data from the standard FB15K and WN18 with Symmetry, Inverse, Anti-symmetry, and Inference patterns. Then, from each subset, we extracted an entirely inductive and a transductive dataset and two more customized datasets. In the first set, the percentage of inductive and transductive triples is fixed, and in the second set, each triplet has one Inductive entity with the other entity being transductive. The description of these settings is in the following.

**Inductive**   We built[3] four datasets with the inductive setting where the entities of test and train datasets are entirely disjoint. We took these fully disjoint sets from both FB15K and WN18 and then subcategorized them into relational datasets, making a set of eight datasets. Table 1 states the statistics of our datasets.

**Transductive**   Transductive Setting has common entities in train and test datasets. Therefore, the entities are already seen by the model, making prediction much easier for them. Table 2 states the statistics of the set of eight datasets from the transductive type.

---

[3]The script to extract data based on each individual relational pattern is available in the code section of
`https://github.com/mlwin-de/relational_pattern_benchmarking/`

Table 2: Transductive Setting Datasets

| Transductive | | Train | Test | Valid | Total Entities | Total Relations | % of Original Dataset |
|---|---|---|---|---|---|---|---|
| FB15K | Symmetry | 5781 | 1399 | 1416 | 2823 | 52 | 1.452 |
| | Anti-Symmetry | 20711 | 128 | 109 | 2471 | 143 | 3.537 |
| | Inverse | 31332 | 750 | 750 | 10988 | 696 | 5.544 |
| | Inference | 70226 | 104 | 111 | 10500 | 377 | 11.895 |
| WN18 | Symmetry | 1449 | 362 | 363 | 2030 | 5 | 1.436 |
| | AntiSymmetry | 6366 | 190 | 168 | 3393 | 15 | 4.441 |
| | Inverse | 4364 | 750 | 750 | 5765 | 17 | 3.873 |
| | Inference | 2027 | 12 | 10 | 3009 | 18 | 1.353 |

Table 3: Head-Tail Inductive Setting Datasets

| Head-Tail Inductive | | Train | Test | Valid | Total Entities | Total Relations | % of Original Dataset |
|---|---|---|---|---|---|---|---|
| FB15K | Symmetry | 5621 | 989 | 990 | 3632 | 52 | 1.283 |
| | Anti-Symmetry | 15404 | 10795 | 10795 | 9241 | 470 | 6.247 |
| | Inverse | 24176 | 4701 | 4701 | 12065 | 794 | 5.670 |
| | Inference | 13845 | 5898 | 5636 | 9671 | 665 | 4.286 |
| WN18 | Symmetry | 1630 | 185 | 186 | 2447 | 5 | 1.322 |
| | AntiSymmetry | 30000 | 5603 | 5603 | 23786 | 18 | 27.217 |
| | Inverse | 5421 | 621 | 621 | 7843 | 17 | 4.401 |
| | Inference | 462 | 84 | 85 | 815 | 15 | 0.417 |

**Head-Tail Ratio Inductive**   We built a set of datasets by keeping either the head or the tail of each triplet in the train hidden from the test dataset. By doing so, we gain a semi-inductive dataset with each triplet unseen. Table 3 reports the statistics of these datasets.

**Percentage-wise building**   In our study, we generated half of the test triples with inductive settings and half with the transductive setting. We apply this percentage base data generation on each category. Table 4 describes the statistics of the 50% datasets.

For our benchmark, we take standard evaluation metrics of Hit Ratios (at 1, 3, 10), Mean Reciprocal Rank, Area Under the Curve, and AUC-PR. Yousef et al. (2008) suggests, for a perfect AUC-PR score, an equal number of negative triplets are needed along with positive triplets. Therefore, as described by Teru et al. (2020), in the test set, the same number of negative samples are created by corrupting the copy of each triplet by either replacing the head or the tail with any random entity. We used the same procedure to incorporate each model with the AUC-PR score in a unified way.
We considered **DistMult** Yang et al. (2015), **RotatE** Sun et al. (2019), **TransE** Bordes et al. (2013), **GraIL** Teru et al. (2020), **MDE** Sadeghi et al. (2020) and **CompGCN** Vashishth et al. (2020) for our analysis.

**Experiment Setup**   Our system is implemented in Python, with Adadelta Zeiler (2012) as the optimizer. All Transductive bias models are set with learning rate **0.0001** with GraIL and CompGCN at **0.01** in-order to practise uniformity. Alpha ($\alpha$) is kept between *[0.5,1]*. The models are set with the Dimensions = *[GraIL = 1000, CompGCN = 100 and all other models = 500]*. The epochs are set uniform for TransE, Distmult and RotatE = **6000** whereas MDE is given a higher number of **150,000** and GraIL and CompGCN are run are **100** and **500** epochs respectively. For TransE, Distmult and RotatE, and MDE, we used a fixed number of negative samples, 50 in all the experiments. To regulate loss function, we estimated the score for 5 runs and took the average. All the experiments are performed on a local server with Intel Corporation Xeon E7 v4/Xeo CPU with 24 cores, 256 GB RAM, and GeForce GTX 1180 with 4 GPU cores.

Table 4: 50% Inductive Setting Datasets

| Percentage Based Inductive | | Train | Test | Valid | Total Entities | Total Relations | % of Original Dataset |
|---|---|---|---|---|---|---|---|
| FB15K | Symmetry | 4677 | 445 | 444 | 3219 | 51 | 0.940 |
| | Anti-Symmetry | 14904 | 11603 | 11608 | 9911 | 472 | 6.436 |
| | Inverse | 6124 | 779 | 779 | 4958 | 600 | 1.297 |
| | Inference | 5840 | 5249 | 5263 | 7031 | 446 | 2.761 |
| WN18 | Symmetry | 2009 | 253 | 253 | 3285 | 5 | 1.661 |
| | AntiSymmetry | 22208 | 5330 | 5329 | 21666 | 18 | 21.709 |
| | Inverse | 7613 | 678 | 678 | 10785 | 17 | 5.924 |
| | Inference | 685 | 58 | 64 | 1235 | 15 | 0.533 |

Table 5: Hit@10 and MRR results of Link Predictors on datasets extracted from FB15K.

| Type of dataset | Dataset | FB15K | | | | | | | | | | | | | |
|---|---|---|---|---|---|---|---|---|---|---|---|---|---|---|---|
| | | DistMult | | TransE | | RotatE | | MDE | | GraIL | | CompGCN | | QuatE | |
| | | Hit@10 | MRR | Hit@10 | MRR | Hit@10 | MRR | Hit@10 | MRR | Hit@10 | MRR | Hit@10 | MRR | Hit@10 | MRR |
| Inductive | Symm | 0.0000 | 0.0003 | 0.0000 | 0.0002 | 0.0000 | 0.0002 | 0.0000 | 0.0002 | 0.0000 | 0.0228 | 0.0074 | 0.0060 | 0.0000 | 0.0040 |
| | Anti-Sym | 0.0000 | 0.0004 | 0.0000 | 0.0002 | 0.0000 | 0.0001 | 0.0000 | 0.0001 | 0.0000 | 0.0236 | 0.0023 | 0.0022 | 0.2435 | 0.0912 |
| | Inverse | 0.0021 | 0.0024 | 0.0028 | 0.0030 | 0.0019 | 0.0021 | 0.0019 | 0.0021 | 0.0000 | 0.0210 | 0.0012 | 0.0013 | 0.2530 | 0.0914 |
| | Inference | 0.0008 | 0.0008 | 0.0018 | 0.0020 | 0.0004 | 0.0005 | 0.0004 | 0.0005 | 0.0000 | 0.0235 | 0.0011 | 0.0013 | 0.2640 | 0.0982 |
| Transductive | Symm | 0.8755 | 0.8692 | 0.1405 | 0.0449 | 0.8594 | 0.8498 | 0.2184 | 0.1232 | 1.0000 | 0.9801 | 0.9836 | 0.8924 | 0.8604 | 0.7914 |
| | Anti-Sym | 0.0041 | 0.0031 | 0.0432 | 0.0155 | 0.0083 | 0.0041 | 0.5078 | 0.3129 | 0.9922 | 0.9836 | 0.9648 | 0.8540 | 0.2214 | 0.1780 |
| | Inverse | 0.0137 | 0.0087 | 0.0439 | 0.0160 | 0.0083 | 0.0069 | 0.1547 | 0.0881 | 0.9953 | 0.9307 | 0.8020 | 0.6305 | 0.2704 | 0.1240 |
| | Inference | 0.0000 | 0.0036 | 0.0800 | 0.0276 | 0.0100 | 0.0066 | 0.1827 | 0.1413 | 0.9932 | 0.9616 | 0.7308 | 0.5264 | 0.1309 | 0.0999 |
| Head/Tail Ratio | Symm | 0.0232 | 0.0097 | 0.0157 | 0.0061 | 0.0071 | 0.0043 | 0.0071 | 0.0047 | 0.4317 | 0.4065 | 0.0137 | 0.0068 | 0.0000 | 0.0166 |
| | Anti-Sym | 0.0019 | 0.0011 | 0.0009 | 0.0008 | 0.0008 | 0.0010 | 0.0084 | 0.0038 | 0.1226 | 0.1143 | 0.0066 | 0.0026 | 0.3997 | 0.1957 |
| | Inverse | 0.0031 | 0.0022 | 0.0341 | 0.0135 | 0.0059 | 0.0039 | 0.0849 | 0.0492 | 0.1380 | 0.1008 | 0.1272 | 0.0734 | 0.3393 | 0.1651 |
| | Inference | 0.0047 | 0.0032 | 0.0141 | 0.0054 | 0.0085 | 0.0046 | 0.0281 | 0.0132 | 0.0882 | 0.0709 | 0.0936 | 0.0583 | 0.3464 | 0.1785 |
| Percentage Based(50%) | Symm | 0.2596 | 0.2590 | 0.0079 | 0.0033 | 0.0461 | 0.0215 | 0.0674 | 0.0415 | 0.2584 | 0.2843 | 0.2629 | 0.2624 | 0.7500 | 0.5461 |
| | Anti-Sym | 0.0002 | 0.0005 | 0.0001 | 0.0003 | 0.0000 | 0.0002 | 0.0006 | 0.0006 | 0.0466 | 0.0422 | 0.0015 | 0.0013 | 0.3738 | 0.2146 |
| | Inverse | 0.0635 | 0.0534 | 0.0161 | 0.0076 | 0.0045 | 0.0028 | 0.0507 | 0.0367 | 0.1343 | 0.1554 | 0.1496 | 0.1310 | 0.2963 | 0.2121 |
| | Inference | 0.0098 | 0.0067 | 0.0142 | 0.0055 | 0.0217 | 0.0110 | 0.1369 | 0.1014 | 0.0782 | 0.0894 | 0.1484 | 0.1291 | 0.1966 | 0.1110 |

Table 6: Hit@10 and MRR results of Link Predictors on datasets extracted from WN18.

| Type of dataset | Dataset | WN18 | | | | | | | | | | | | | |
|---|---|---|---|---|---|---|---|---|---|---|---|---|---|---|---|
| | | DistMult | | TransE | | RotatE | | MDE | | GraIL | | CompGCN | | QuatE | |
| | | Hit@10 | MRR | Hit@10 | MRR | Hit@10 | MRR | Hit@10 | MRR | Hit@10 | MRR | Hit@10 | MRR | Hit@10 | MRR |
| Inductive | Symm | 0.0000 | 0.0001 | 0.0000 | 0.0002 | 0.0000 | 0.0001 | 0.0000 | 0.0001 | 0.0000 | 0.0201 | 0.0018 | 0.0011 | 0.0000 | 0.0000 |
| | Anti-Sym | 0.0000 | 0.0002 | 0.0000 | 0.0001 | 0.0000 | 0.0001 | 0.0000 | 0.0001 | 0.0001 | 0.0304 | 0.0006 | 0.0007 | 0.0000 | 0.0016 |
| | Inverse | 0.0000 | 0.0001 | 0.0000 | 0.0001 | 0.0000 | 0.0001 | 0.0000 | 0.0001 | 0.0000 | 0.0215 | 0.0000 | 0.0003 | 0.0000 | 0.0009 |
| | Inference | 0.0000 | 0.0002 | 0.0000 | 0.0002 | 0.0000 | 0.0003 | 0.0000 | 0.0004 | 0.0000 | 0.0251 | 0.0051 | 0.0042 | 0.0000 | 0.0002 |
| Transductive | Symm | 0.9277 | 0.9064 | 0.0073 | 0.0031 | 0.7509 | 0.9792 | 0.4378 | 0.4195 | 1.0000 | 0.9979 | 0.9848 | 0.9738 | 0.9143 | 0.9466 |
| | Anti-Sym | 0.0000 | 0.0005 | 0.0000 | 0.0016 | 0.0000 | 0.0006 | 0.1079 | 0.0624 | 0.9684 | 0.9581 | 0.9816 | 0.9720 | 0.0374 | 0.0153 |
| | Inverse | 0.0000 | 0.0003 | 0.0000 | 0.0020 | 0.0047 | 0.0030 | 0.2220 | 0.1787 | 1.0000 | 0.9977 | 0.9940 | 0.9441 | 0.0377 | 0.0156 |
| | Inference | 0.2917 | 0.2394 | 0.0000 | 0.0013 | 0.0833 | 0.0523 | 0.2500 | 0.2520 | 1.0000 | 1.0000 | 0.2083 | 0.1566 | 0.0486 | 0.0323 |
| Head/Tail Ratio | Symm | 0.0027 | 0.0008 | 0.0000 | 0.0002 | 0.0055 | 0.0013 | 0.0162 | 0.0111 | 0.0054 | 0.0253 | 0.0081 | 0.0063 | 0.0272 | 0.0089 |
| | Anti-Sym | 0.0007 | 0.0005 | 0.0000 | 0.0005 | 0.0014 | 0.0014 | 0.0003 | 0.0003 | 0.0011 | 0.0291 | 0.0012 | 0.0006 | 0.0844 | 0.0453 |
| | Inverse | 0.0009 | 0.0007 | 0.0000 | 0.0004 | 0.0016 | 0.0015 | 0.0523 | 0.0318 | 0.0000 | 0.0340 | 0.0395 | 0.0196 | 0.0688 | 0.0577 |
| | Inference | 0.0000 | 0.0020 | 0.0000 | 0.0002 | 0.0000 | 0.0002 | 0.0833 | 0.0277 | 0.0000 | 0.0200 | 0.0714 | 0.0321 | 0.0198 | 0.0245 |
| Percentage Based(50%) | Symm | 0.5013 | 0.5029 | 0.0000 | 0.0003 | 0.0731 | 0.0437 | 0.0514 | 0.0353 | 0.5020 | 0.5119 | 0.0040 | 0.0030 | 0.8593 | 0.7899 |
| | Anti-Sym | 0.0000 | 0.0002 | 0.0002 | 0.0002 | 0.0000 | 0.0001 | 0.0005 | 0.0006 | 0.0111 | 0.0439 | 0.0010 | 0.0010 | 0.0198 | 0.0245 |
| | Inverse | 0.1924 | 0.1199 | 0.0118 | 0.0047 | 0.0701 | 0.0368 | 0.0155 | 0.0082 | 0.4808 | 0.4906 | 0.0000 | 0.0010 | 0.0193 | 0.0102 |
| | Inference | 0.0000 | 0.0002 | 0.0000 | 0.0005 | 0.0000 | 0.0007 | 0.0172 | 0.0174 | 0.0345 | 0.0552 | 0.0086 | 0.0121 | 0.1086 | 0.0196 |

# 5 Results

In this Section, we report the experiment results and discuss them. Tables 5 and 6 show the MRR and Hit@10 performance of the LP methods and the Tables 7, 8 and 9 report the AUC-PR results. Extended result sheets are available in `https://github.com/mlwin-de/relational_pattern_benchmarking`.

**Inductive**  As far as Inductive datasets are concerned, in both FB15K and WN18, GraIL and CompGCN outperform all other models due to their property of inductive bias-ness. Segregating far, all models performed well over the Inference dataset. CompGCN gave better performance on Inverse dataset with a 56.46 AUC-PR score on the FB15K extracted dataset and 67.40 AUC-PR score on the WN18 extracted dataset, whereas MDE did not perform well in all datasets from the inductive category. To sum up, the difference between Inductive and Transductive models could easily be noticed by this set of experiments.

Table 8 summarizes FB15K results and the green column of Table 7 summarizes the average AUC-PR on the Inductive setting.

All models show a low performance on the MRR and Hit metric for the Inductive datasets as in the Tables 5 and 6. However, the AUC-PR results show that some models can better distinguish positive and negative samples in an equal number of samples, where CompGCN shows a better performance in separating negative samples in the Inductive setting.

**Transductive**  For evaluation with transductive setting of datasets, all models showed great improvement in link prediction task than their performance over other datasets. TransE and MDE was almost 15% - 20% less accurate as compared to other state of the art models in the AUC-PR. Symmetry dataset gives promising results with link prediction of more than 95% in almost all models. Furthermore, for WN18 dataset, GraIL and even CompGCN predicts 99% true triplets for the symmetry dataset. Overall GraIL and CompGCN and QuatE are proved to be the superior models for transductive datasets.

Table 7: Mean AUC-PR performance of the LP methods.

| Type of dataset | WN18 | | | | FB15K | | | |
|---|---|---|---|---|---|---|---|---|
| | Metric (AUC-PR) | | | | | | | |
| | Inductive | Transductive | Head/Tail Ratio | Percentage Based(50%) | Inductive | Transductive | Head/Tail Ratio | Percentage Based(50%) |
| DistMult | 0.5015 | 0.7203 | 0.5456 | 0.6634 | 0.4995 | 0.6255 | 0.5468 | 0.5755 |
| TransE | 0.4688 | 0.6239 | 0.5169 | 0.5643 | 0.4489 | 0.6685 | 0.5969 | 0.5266 |
| RotatE | 0.4542 | 0.7497 | 0.5840 | 0.6231 | 0.4282 | 0.6697 | 0.5829 | 0.5336 |
| MDE | 0.4865 | 0.9624 | 0.6426 | 0.5952 | 0.4474 | 0.8709 | 0.5937 | 0.5310 |
| GraIL | 0.5000 | 0.9960 | 0.5013 | 0.6433 | 0.5004 | 0.9975 | 0.6105 | 0.5926 |
| CompGCN | 0.5531 | 0.9269 | 0.5713 | 0.5596 | 0.6368 | 0.9976 | 0.6473 | 0.6768 |
| QuatE | 0.4426 | 0.6518 | 0.6518 | 0.7382 | 0.5867 | 0.9618 | 0.6563 | 0.8208 |

Table 8: AUC-PR Results of Link Predictors on datasets extracted from FB15K.

| Type of dataset | DataSets | FB15K | | | | | | |
|---|---|---|---|---|---|---|---|---|
| | | Metric (AUC-PR) | | | | | | |
| | | DistMult | TransE | RotatE | MDE | GraIL | CompGCN | QuatE |
| Inductive | Symm | 0.4933 | 0.4585 | 0.4554 | 0.4650 | 0.5000 | 0.7682 | 0.3115 |
| | Anti-Symmetry | 0.4993 | 0.4161 | 0.3889 | 0.4221 | 0.5009 | 0.6577 | 0.4841 |
| | Inverse | 0.4992 | 0.4632 | 0.4282 | 0.4496 | 0.4986 | 0.5646 | 0.4824 |
| | Inference | 0.5063 | 0.4578 | 0.4403 | 0.4527 | 0.5022 | 0.5566 | 0.4924 |
| Transductive | Symm | 0.9618 | 0.6901 | 0.9608 | 0.9434 | 0.9966 | 0.9995 | 1.0000 |
| | Anti-Symmetry | 0.5151 | 0.6038 | 0.5853 | 0.9986 | 0.9998 | 1.0000 | 0.8998 |
| | Inverse | 0.5345 | 0.6318 | 0.5967 | 0.7948 | 0.9963 | 0.9927 | 0.9137 |
| | Inference | 0.4907 | 0.7484 | 0.5359 | 0.7468 | 0.9971 | 0.9982 | 0.8933 |
| Head/Tail Ratio | Symm | 0.6409 | 0.7480 | 0.7259 | 0.7764 | 0.7846 | 0.6109 | 0.6526 |
| | Anti-Symmetry | 0.5054 | 0.5486 | 0.5406 | 0.5331 | 0.5881 | 0.5559 | 0.6539 |
| | Inverse | 0.5093 | 0.5507 | 0.5359 | 0.5499 | 0.5432 | 0.7815 | 0.6906 |
| | Inference | 0.5314 | 0.5401 | 0.5292 | 0.5156 | 0.5261 | 0.6410 | 0.6102 |
| Percentage Based(50%) | Symm | 0.7257 | 0.5955 | 0.6256 | 0.6201 | 0.6831 | 0.7591 | 0.7721 |
| | Anti-Symmetry | 0.4959 | 0.4474 | 0.4347 | 0.4208 | 0.5045 | 0.5774 | 0.7268 |
| | Inverse | 0.5870 | 0.5351 | 0.5602 | 0.5597 | 0.6206 | 0.7253 | 0.7368 |
| | Inference | 0.4935 | 0.5284 | 0.5140 | 0.5235 | 0.5620 | 0.6455 | 0.7169 |

An exception of performance for Symmetry dataset in Distmult between inductive and transductive case was observed, where for both WN18 and FB15K the AUC-PR result increased in the transductive setting by an amount about 40%.

Promising AUC-PR results for symmetry dataset by the model are displayed in Table 8 and Table 9.

**Semi-Inductive - Head Tail Ratio** Since this set of datasets is also inductive in its properties as all triplets are fully inductive with either one of the entities unseen. The models behave exactly the same way as they work with the inductive datasets. All state-of-the-art models fail to perform under such settings except GraIL and CompGCN due to their inductive nature. Despite the fact that GraIL and CompGCN perform better than transductive bias model, they could not rank the triplets correctly with more than 0.52 AUC-PR. However, CompGCN shows some improvements for the Inference type datasets under the category with around 0.60 AUC-PR.

**Semi-Inductive - Percentage-Wise** The datasets are gathered on the principle of half datasets from the inductive category with 50% triplets of the transductive nature. Due to the reason, they perform well on the link prediction models. Distmult and CompGCN gives promising results of more than 85% on the symmetry datasets. TransE, due to its inability to infer symmetric patterns could not give better score but predicts inference patterns better than any other. As an average, we conclude that all models outperform over the transductive setting of datasets in link prediction task. The KGE models also give promising results over datasets with half inductive and half transductive type (Count-Based Dataset), evidently due to the fact of containing exactly 50% transductive triplets. Inductive datasets and head-tail inductive datasets still faces issues in link prediction task with better results on CompGCN and GraIL.

## 5.1 Discussion

**Inductive Evaluations** Since none of the three types of inductive experiments included known triples in the test phase, the models did not use any other information indicative of the identity of tested entities (e.g., their distance to other entities), except for GraIL that applies sub-graph calculations for

Table 9: AUC-PR Results of Link Predictors on datasets extracted from WN18.

| Type of Dataset | DataSets | WN18 Metric (AUC-PR) | | | | | | |
| --- | --- | --- | --- | --- | --- | --- | --- | --- |
| | | DistMult | TransE | RotatE | MDE | GraIL | CompGCN | QuatE |
| **Inductive** | **Symm** | 0.4912 | 0.4864 | 0.5164 | 0.4889 | 0.5000 | 0.4736 | 0.9892 |
| | **Anti-Symmetry** | 0.4972 | 0.4400 | 0.4119 | 0.4774 | 0.4999 | 0.6740 | 0.4616 |
| | **Inverse** | 0.5023 | 0.4312 | 0.3931 | 0.4667 | 0.5000 | 0.5248 | 0.4716 |
| | **Inference** | 0.5154 | 0.5175 | 0.4953 | 0.5131 | 0.5000 | 0.5398 | 0.4242 |
| **Transductive** | **Symm** | 0.9855 | 0.6506 | 0.9849 | 0.9395 | 1.0000 | 0.9993 | 1.0000 |
| | **Anti-Symmetry** | 0.5705 | 0.6048 | 0.5183 | 0.9856 | 0.9842 | 0.9953 | 0.9875 |
| | **Inverse** | 0.4507 | 0.5274 | 0.6299 | 0.9244 | 1.0000 | 0.9952 | 0.9683 |
| | **Inference** | 0.8745 | 0.7127 | 0.8655 | 1.0000 | 1.0000 | 0.7179 | 0.8913 |
| **Head/Tail Ratio** | **Symm** | 0.5390 | 0.5609 | 0.6477 | 0.6652 | 0.5053 | 0.6172 | 0.6449 |
| | **Anti-Symmetry** | 0.5328 | 0.5139 | 0.5135 | 0.4969 | 0.4999 | 0.5722 | 0.6258 |
| | **Inverse** | 0.5202 | 0.5746 | 0.5442 | 0.6060 | 0.5000 | 0.5812 | 0.7249 |
| | **Inference** | 0.5905 | 0.4181 | 0.6306 | 0.8024 | 0.5000 | 0.5145 | 0.6294 |
| **Percentage Based(50%)** | **Symm** | 0.8185 | 0.5920 | 0.7657 | 0.7487 | 0.7626 | 0.5888 | 0.7873 |
| | **Anti-Symmetry** | 0.4942 | 0.4773 | 0.4836 | 0.4769 | 0.5103 | 0.5973 | 0.8900 |
| | **Inverse** | 0.7936 | 0.6586 | 0.7142 | 0.6418 | 0.7974 | 0.5374 | 0.8134 |
| | **Inference** | 0.5472 | 0.5291 | 0.5290 | 0.5134 | 0.5031 | 0.5150 | 0.7924 |

the test-set as well. Thus, the vector weights of the unknown entities are set randomly. Consequently, we conclude that the experimented models that benefit from no knowledge of entities, estimate the rank of the samples only by generating an embedding space such that the scores of negative triples are separable from the random values to some extent using the common relations between test and the train dataset. In the following, we generally compare the performance of models over by datasets extracted from the two original knowledge graphs.

**FB15K**   For FB15K, GraIL and CompGCN perform well on the inductive datasets with the highest performance on the Symmetry dataset in CompGCN whereas GraIL shows a near performance on all datasets with the inductive setting. CompGCN also performs well on transductive datasets with almost 100% accuracy on Anti-Symmetry along with outstanding performance on all datasets with the Count-based setting. QuatE and GraIL show exactly the same trend as CompGCN with near accuracy. MDE performs well on transductive setting datasets with 99% accuracy on the Anti-Symmetry followed by Head-Tail Ratio based setting datasets. RotatE, TransE and DistMult showed the same performance pattern as MDE with the highest performance in the transductive setting and least in inductively set datasets. Table 8 describes the result of these evaluations.

**WN18**   In datasets extracted from WN18 is concerned, a similar pattern as the FB15K dataset is observable. GraIL and CompGCN show better results on inductive settings as compared to other models. Link prediction in the inductive and the transductive setting is better on WN18 datasets in most of the models as compared to FB15K that evidently is due to the fact of fewer relations to be computed. Link prediction on Symmetry datasets from the transductive setting has around 0.99 AUC-PR for almost all the models considered. Table 9 quotes results on WN18 Datasets.

## 5.2   Evaluation of GFA-NN model

Finally, we benchmark our datasets over a newer multi-objective optimization KGE similar to MDE, i.e., GFA-NN Sadeghi et al. (2021a), which considers the datasets' graphical features to create embeddings. The node features and graph properties are calculated and stored in separate files for training purposes. The graphical feature information is not available to the model in the evaluation phase to make a fair comparison.

Comparison between the previous MDE and GFA-NN reported in Table 10 shows outstanding behavior of the newly introduced method in the Transductive and Semi-inductive settings. The most significant improvement is observable in the Inference datasets, where the results show an improvement of between 0.6% to 22% in the tested cases. For Instance, the FB15K Inference dataset with the transductive setting improved from 74% in the AUC-PR score to 96%. Lastly, the results in the purely inductive setting only shows an improvement in the Inference dataset, that is because we handicapped GFA-NN, not to know the graphical features of any nodes in the test dataset.

Table 10: Evaluation results of GFA-NN model compared to MDE.

| LP Methods | Type of DataSet | WN18 | | | Metric (AUC-PR) | FB15K | | | |
|---|---|---|---|---|---|---|---|---|---|
| | | Symm | Anti-Symmetry | Inverse | Inference | Symm | Anti-Symmetry | Inverse | Inference |
| MDE | Inductive | 0.4889 | 0.4774 | 0.4667 | 0.5131 | 0.4650 | 0.4221 | 0.4496 | 0.4527 |
| | Transductive | 0.9395 | 0.9856 | 0.9244 | 1.0000 | 0.9434 | 0.9986 | 0.7948 | 0.7468 |
| | Head/Tail Ratio | 0.6652 | 0.4969 | 0.6060 | 0.8024 | 0.7764 | 0.5331 | 0.5499 | 0.5156 |
| | Percentage Based(50%) | 0.7487 | 0.4769 | 0.6418 | 0.5134 | 0.6201 | 0.4208 | 0.5597 | 0.5235 |
| GFA-NN | Inductive | 0.4572 | 0.4218 | 0.3861 | 0.4877 | 0.4593 | 0.3939 | 0.4477 | 0.4584 |
| | Transductive | 1.0000 | 0.9995 | 1.0000 | 1.0000 | 1.0000 | 0.9999 | 0.9848 | 0.9695 |
| | Head/Tail Ratio | 0.6653 | 0.4977 | 0.6409 | 0.5846 | 0.6289 | 0.5470 | 0.6425 | 0.5764 |
| | Percentage Based(50%) | 0.7600 | 0.4662 | 0.7301 | 0.5382 | 0.6842 | 0.4264 | 0.6024 | 0.5437 |

Table 11: Ranking results of the LP models on the aggregate datasets.

| Model | WN18RR | | | FB15k-237 | | |
|---|---|---|---|---|---|---|
| | MR | MRR | Hit@10 | MR | MRR | Hit@10 |
| QuatE | – | 0.482 | 0.572 | – | **0.366** | **0.556** |
| TransE | 357 | 0.294 | 0.501 | 357 | 0.294 | 0.465 |
| DistMult | 5261 | 0.44 | 0.49 | 254 | 0.241 | 0.419 |
| CompGCN | 3533 | 0.479 | 0.546 | 197 | 0.355 | 0.535 |
| RotatE | 3340 | 0.476 | 0.571 | 177 | 0.338 | 0.533 |
| MDE | 3219 | 0.458 | 0.536 | 203 | 0.344 | 0.531 |
| GFA-NN | 3390 | **0.486** | **0.575** | 186 | 0.338 | 0.522 |

## 5.3 Comparison to Aggregate datasets

Table 11 shows the results of the LP methods on the aggregate datasets reported in Sadeghi et al. (2021a) and Sun et al. (2019).

This comparison of this table results to our pattern-specific experiments shows that while the difficulty of each individual relation pattern for each model influences the overall results, the ratio of each relation pattern also impacts their performance because the results are averaged in the MRR and Hit evaluations. For example, while CompGCN is not the best method in learning inverse relations in the transductive setting, it is one of the most efficient methods in the FB15k-237 because FB15k-237 and WN18RR are missing the inverse relations from FB15K and WN18.

## 6 Conclusion

To sum up, we created several standard datasets with distinct relational patterns ranging from symmetry to inverse and different degrees of inductiveness. We evaluated several state-of-the-art Link Prediction models over them. We extended our research in working on link prediction models by this benchmarking approach and highlighting their working on different types of datasets. A unified evaluation strategy of AUC-PR measurement is incorporated into all link prediction models beside the Hit@K and MRR measures. We highlighted the datasets with improved performance on particular LP models. In order to support further research in the domain, we incorporated our benchmarking datasets to the BenchEmbedd for evaluation of a linked data life-cycle. The meaningful experiments outcome indicate that these datasets will foster the research on Link Prediction and KG embeddings.

## Acknowledgements

We acknowledge the support of the following projects: SPEAKER (BMWi FKZ 01MK20011A), JOSEPH (Fraunhofer Zukunftsstiftung), the EU projects Cleopatra (GA 812997), TAILOR (EU GA 952215), the BMBF project MLwin (01IS18050) and the BMBF excellence clusters ML2R (BmBF FKZ 01 15 18038 A/B/C) and ScaDS.AI (IS18026A-F).

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
