# Supplementary Material

Afshin Sadeghi[1,2], Hirra Abdul Malik[1], Diego Collarana[2,3], and Jens Lehmann[1,2]

[1]SDA Research Group, University of Bonn, Germany
[2]Fraunhofer IAIS, Dresden, Germany
[3]Universidad Privada Boliviana, Bolivia
*{s6hiabdu,lehmann}*.uni-bonn.de,
*{afshin.sadeghi,diego.collarana.vargas,jens.lehmann}@iais.fraunhofer.de*

Hereby the authors confirm that,

- Link to access the dataset and its metadata is available: `https://github.com/mlwin-de/relational_pattern_benchmarking`.

- The dataset is consist of n-triples which is an open and widely used data format for evaluating embedding models. We provided a detailed explanation on how the dataset can be read on the home page of the dataset.

- The authors ensure Long-term preservation of the dataset by maintaining it on the github website as the repository.

- Authors use CC licence and an Explicit CC license is included in the dataset folder. Authors bear all responsibility in case of violation of rights.

- Authors have added structured metadata to a dataset's meta-data page using RDFa in the `https://github.com/mlwin-de/relational_pattern_benchmarking/Dataset/info-rdfa.htm` and a DOI is provided to the repository.

- For the benchmark over the datasets, all the hyperparameters required to reproduce the results are available and documented in the github page of the dataset. In addition, we extended BenchEmbedd environment to benchmark future models based on FAIR principles.

  Dataset documentation and intended uses. Recommended documentation frameworks include datasheets for datasets, dataset nutrition labels, data statements for NLP, and accountability frameworks.

35th Conference on Neural Information Processing Systems (NeurIPS 2021) Track on Datasets and Benchmarks.