# OpenReview forum: "Relational Pattern Benchmarking on the Knowledge Graph Link Prediction Task"
_NeurIPS.cc/2021/Track/Datasets_and_Benchmarks/Round2 — NeurIPS 2021 Datasets and Benchmarks Track (Round 2)_

### Official Review · Reviewer_zaK2 · 2021-09-14
**Benchmarking link prediction on KG with novel formulation**

**Rating:** 7
**Confidence:** 3
**Correctness:** The data collection process and the e…
**Clarity:** The paper is well written and clearly…

**Strengths:**

1. A novel perspective of formatting link prediction tasks on KG;
2. Evaluation pipeline is well-defined;
3. The comparisons are rigorous with detailed analysis;

**Weaknesses:**

1. The sub-datasets are evaluated, however, the evaluation of the whole KG might also be interested. more importantly, what's the relation between the performance of the subsets and the whole KG?;

**Additional Feedback:**

1. Will the pattern formulations of links in KG motivate new prospectives of solving reasoning tasks (and other tasks) on KG?

**Documentation:**

The dataset creation process is reasonably well documented.

**Ethics:**

I do not have ethical concerns.

**Relation To Prior Work:**

Existing datasets and their limitations are discussed. To the best of my knowledge, the pattern formulation is novel.

**Summary And Contributions:**

This work creates two datasets for link prediction on KG with novel and motivating pattern formulations. With the defined patterns, each dataset could be divided into 16 sub-datasets. The evaluation strategy is clearly stated and convincing. With the benchmarking results comes detailed analysis.

---

> ### Author Response · Authors · 2021-09-30
> **Relation to the aggregate Datasets.**
>
> Thanks for pointing out this interesting question.  We included a table for aggregate results and added a discussion about the mentioned question.
>
> While the difficulty of each individual relation pattern for each model influences the overall results, the ratio of each relation pattern also impacts their performance because the results are averaged in the MRR and Hit evaluations.
> For example, while CompGCN is not the best method in learning inverse relations, it is one of the most efficient methods in the FB15k-237 because FB15k-237 and WN18RR are missing the inverse relations from FB15K and WN18.

---

### Official Review · Reviewer_UL95 · 2021-09-19
**A dataset exploring LP model's ability of different relation patterns.**

**Rating:** 6
**Confidence:** 3

**Strengths:**

Adequate properties considerations. The author constructed four categories:  Fully Inductive, Fully Transductive, CountBased Inductive, and either Head or Tail Inductive. Each category is further divided into patterns of Symmetry, Anti-symmetry, Inverse and Inductive, making the evaluation comprehensive.

**Weaknesses:**

Some patterns have very little data. I'm afraid the results may be influenced by the data bias.

**Additional Feedback:**

No

**Clarity:**

From a personal point of view, I suggest the author to refer Table 5 in Page 6 Inductive part or put Table 5 before the Figure 3.  It will make the results clearer. It is also better to use a new section to illustrate the results rather than put them in Section 4.1.

**Correctness:**

In section 3.1.4, the example is not Figure2d and should be Figure 2e. Meanwhile, the figure2d seems to be not a composition relation?

**Documentation:**

The author provided the github repository which contains the required documents. It also shows sufficient detail on data collection and organization, availability and maintenance.

**Ethics:**

This paper has no ethics issues.

**Relation To Prior Work:**

Previous work to evaluate the LP models' ability are usually too general and cannot analyze the advantages and disadvantages of the model in more detail . This work explores specific patterns and types of dataset to better analyze the performance of the model.

**Summary And Contributions:**

This paper builds several specific Link Prediction datasets and group them into different categories based on the relation patterns and properties. The paper evaluates  state-of-the-art KG embedding models and the results are informative and directive.

---

> ### Author Response · Authors · 2021-09-30
> **Thanks for the precise analysis of the paper structure.**
>
> Thanks for the comment about data size. We considered this challenge when extracting the datasets by manually checking for outliers to avoid biased cases.
> To reflect your precise analysis of the figures and tables, we updated them according to the comments and structured all the results and discussions in one new section.

---

### Official Review · Reviewer_adwM · 2021-09-21
**Benchmarking dataset based on FB15k and WN18**

**Rating:** 4
**Confidence:** 4

**Strengths:**

The paper introduces several subtasks of two common datasets, that could potentially be highly useful in identifying the specific strengths and weaknesses of different models. The authors evaluate several common as well as recent models from the literature on each subtask. This leads to interesting observations, e.g. a surprising resilience of DistMult compared to other embedding-based approaches (RotatE, TransE) to the move to an inductive setting in Figure 3.

**Weaknesses:**

This benchmark uses subsets of the FB15k and WN18 datasets. There are well-known issues of data leakage in these, discussed e.g. for FB15k in [1]. Essentially, some relations are encoded multiple times through e.g. explicit inverses (such as (x, father_of, y) and (y, inverse-father_of, x), with occurrences of the same relation in train and test sets because of this. Corrected versions of the datasets are available in the form of FB15k-237 [1], and WN18-RR [2]. The authors should either use the corrected datasets, or ensure the removal of any potential leakage from their dataset -- the latter can be tested through the NodeFeat and LinkFeat baselines from [1].

A second, lesser issue is the choice of evaluation metric. FB15k and WN18 are typically evaluated through ranking metrics (MRR, hit ratios), while this paper employs AUC-PR. This makes comparisons to older numbers on the full datasets difficult.

[1] Toutanova and Chen, 2015. Observed versus latent features for knowledge base and text inference. CVSC.

[2] Dettmers et al., 2018. Convolutional 2D Knowledge Graph Embeddings. AAAI.

**Additional Feedback:**

In Table 7 and section 4.3, a comparison between two models on the introduced subsets of FB15k is made. Given that this benchmark includes both FB15k and WN18, it is a bit strange that an equivalent comparison for WN18 is not included.

**Clarity:**

The paper is moderately well-written, but there are issues of clarity when it comes to the reporting of numbers. Some baseline performances are presented through bar diagrams, which are difficult to read -- the numbers are small, and the scale of the axes varies from diagram to diagram. This makes it difficult to understand the relative performance of each strategy in each setting. As far as I can see the aggregate performances for the inductive/transductive settings are presented as bar diagrams, with individual performances on each task presented in a table; it is not clear why this choice was made, and clarity would benefit immensely from moving the aggregates to a table format as well.

**Correctness:**

Unfortunately, the dataset construction is somewhat problematic; see my comments on weaknesses.

**Documentation:**

Sufficient detail is included.

**Ethics:**

No ethical concerns come to mind.

**Relation To Prior Work:**

The paper should discuss and relate to work identifying problems with the underlying data; see my comments on weaknesses.

**Summary And Contributions:**

This paper introduces a new benchmark for knowledge base link prediction based on the FB15k and WN18 datasets. The authors identified several types of reasoning models must employ to solve these datasets, and selected subsets that represent these separate problems. Each subset is to be used in an inductive and a transductive setting. In the inductive setting, the entities of the training and test sets are kept entirely separate; in the transductive setting, entities from the training set can reoccur in the test set. The authors evaluate several models from the literature on each subset.

---

> ### Author Response · Authors · 2021-09-30
> **Thanks for pointing out the importance of the Hit and MMR metrics.**
>
> Thank you for this important remark about inverse-triples of train samples occurring in the test dataset. In our check, 2 of the datasets from the 28 are affected by this issue, not the complete dataset. Moreover, based on our calculations the impact in the results is really low, i.e., it will influence just the results of the FB15K/Transductive/inference and AntiSym by two percent. As future work, we plan a second version of the dataset removing these issues of the original dataset.
>
> Thanks for pointing out the importance of the Hit and MMR metrics in our table reports. The limited space made us keep only AUC-PR results, which produce more meaningful comparisons on the inductive datasets. However, the MRR, Hit results, and the WN18 tests over GPA-NN, are included on the dataset Github page. We brought them back to the paper as suggested. We as well replaced bar diagrams with tables to improve the clarity.

---

### Official Review · Reviewer_soML · 2021-09-27
**A fine-grained benchmark for knowledge graph link prediction**

**Rating:** 7
**Confidence:** 3
**Clarity:** The paper is clearly written and easy…

**Strengths:**

1. This is the first benchmark that highlights the different reasoning patterns in KG link prediction.
2. The benchmark and the results present here can help future identify the remaining challenges in KB link prediction

**Weaknesses:**

The benchmark is mostly based on existing dataset but only provides additional information for each instance (e.g., whether it's inductive).

**Additional Feedback:**

N/A

**Correctness:**

The benchmark construction looks standard and sound. The baselines are standard and reasonable for provide some initial insights.

**Documentation:**

yes

**Ethics:**

I don't seen any ethical issue.s

**Relation To Prior Work:**

The authors might consider discussing more work related to KB reasoning, e.g., methods than can handle more complex queries.

**Summary And Contributions:**

This paper presents a KG benchmark with subsets of different patterns (e.g., inductive, transductive). This is in contrast to existing benchmarks that do not separate the KG triples. While the dataset is built upon existing link prediction datasets, this benchmark can be useful to get new insights about existing link prediction methods. In additional, this work also includes baseline experiments and initial findings on different relational patterns, e.g, the distinctive performance between inductive and transductive subsets.

---

> ### Author Response · Authors · 2021-09-30
> **Thanks for pointing out the relation to KB reasoning works.**
>
> Thanks for the very helpful comments. About the related work, we enriched it by adding two recent works: Yang et al. (2017) and Meilicke et al. (2018) that construct complex datasets for logical inference and reasoning tasks. However, in our dataset, we included the most frequent graph patterns.
>
> ​ Yang et al. (2017): Yang, F., Yang, Z., and Cohen, W. W.. Differentiable learning of logical rules for knowledge
> base reasoning.
>
>  Meilicke et al. (2018) Meilicke, C., Fink, M., Wang, Y., Ruffinelli, D., Gemulla, R., and Stuckenschmidt, H. . Fine-grained evaluation of rule-and embedding-based systems for knowledge graph completion.

---

### Public Comment · ~Ralph_Abboud1 · 2021-09-29
**Related Work**

Hi, I'm writing to point out a very closely related work of BoxE (Abboud et al., 2020) which is completely missed in this work. Inference patterns are comprehensively studied in the BoxE paper, including patterns such as symmetry, anti-symmetry, inversion, and relational hierarchies. These patterns are even studied jointly and BoxE is formally shown to capture a rich rule language. Furthermore, an empirical evaluation is given for rule injection on a dedicated dataset.

---

> ### Author Response · Authors · 2021-09-30
> **Related work**
>
> Thanks for reaching out and mentioning the model. We did not include the work in the related work since we focused on dataset and benchmark papers. However, we would like to extend the study to include more models in the benchmark and to maintain the github page with the new results (https://github.com/mlwin-de/relational_pattern_benchmarking). So please keep in touch.

---

### Decision · Program_Chairs · 2021-10-09

**Decision:**

Accept

**Comment:**

This paper proposes to split existing datasets to subsets such that we can test whether models for knowledge prediction behave in a way that respects various reasoning properties. All reviewers agree this dataset is useful and the results are interesting and illuminating. Several concerns have been raised that must be addressed in the final version: (a) using versions of the datasets where there is no leakage from train to test, though authors explain this might affect just a small portion of the data (b) improve clarity (c) improve related work and refer to models that try to specifically look at properties such as symmetry and explain how the evaluation here is more comprehensive than prior work.